# Water, sanitation, and depressive symptoms in Indonesia: The mediating role of life satisfaction

**Mashita Fajri**[1], **Sri Idaiani**[2], **Amy Blakemore**[3], **Jonathan Gibson**[4], **Herni Susanti**[1], **Helen Brooks**[3], **Penny Bee**[3], **Asri Maharani**[3]*

**1** Department of Mental Health Nursing, Faculty of Nursing, Universitas Indonesia, Depok, Indonesia, **2** Research Centre for Preclinical and Clinical Medicine, National Research and Innovation Agency, Cibinong, Indonesia, **3** Division of Nursing, Midwifery & Social Work, University of Manchester, Manchester, United Kingdom, **4** Centre for Primary Care and Health Services Research, University of Manchester, Manchester, United Kingdom

* asri.maharani@manchester.ac.uk

## Abstract

Access to clean water and adequate sanitation is vital for public health. However, the effects of water and sanitation on depressive symptoms remain insufficiently explored among the general population in low- and middle-income countries (LMICs). This study aims to examine the links between sanitation conditions and depressive symptoms, and to evaluate the mediating role of life satisfaction (LS) among adults in Indonesia. We used the fifth wave of the Indonesian Family Life Survey (IFLS-5) dataset with participants aged 15 and above (mean: 37.32 years; SD: 14.93), collected in September 2014/March 2015. Depressive symptoms were measured using the CES-D-10 scale. Five sanitation indicators were included: drinking water, water source, toilet facilities, liquid waste disposal and safe waste disposal methods. Structural equation modelling (SEM) was employed to estimate the direct and indirect pathways linking sanitation factors and depressive symptoms through LS, adjusting for age and sex. The study involved 31,446 participants, of whom 7,312 (23%) were classified as having depressive symptoms. In the first SEM model, drinking water ($\beta = 0.023$, $p < 0.001$), water source ($\beta = 0.033$, $p < 0.001$), toilet facilities ($\beta = 0.039$, $p < 0.001$), sewage disposal ($\beta = 0.027$, $p < 0.001$) and waste disposal method ($\beta = 0.021$, $p < 0.005$) were directly associated with depressive symptoms. In the second SEM model, which included LS as a mediator, the direct effects remained consistent: drinking water ($\beta = 0.013$, $p < 0.05$), water source ($\beta = 0.054$, $p < 0.001$), toilet facilities ($\beta = 0.068$, $p < 0.001$), sewage disposal ($\beta = 0.022$, $p < 0.001$) and waste disposal method ($\beta = 0.046$, $p < 0.001$). Additionally, unimproved sanitation was significantly linked to lower LS and LS was strongly associated with depressive symptoms across all sanitation factors ($\beta \approx 0.124$–$0.126$, $p < 0.001$). Poor water and sanitation are associated with a higher risk of depressive symptoms among Indonesian adults, with LS as a partial mediator of this relationship. These findings highlight the need to

**Data availability statement:** We would like to draw your attention that the paper is based on a secondary data analysis of the Indonesia Family Life Survey (IFLS), which is publicly available for download after registration with the RAND corporation/IFLS website. As per their policy, the database is freely accessible only to those researchers who have registered their interest on the RAND website. We have provided the link to download the data upon registration in our paper. The IFLS-5 data are open for public use after registration on RAND Corporation website (http://www.rand.org/labor/FLS/IFLS/ifls5.html).

**Funding:** This research was funded by the National Institute for Health and Care Research (NIHR) [Global Health Research for Sustainable Care for Anxiety and Depression in Indonesia (Award ID NIHR134638)] using UK international development funding from the UK Government to support global health research. The views expressed in this publication are those of the author(s) and not necessarily those of the NIHR or the UK Government. The funders had no role in the study design, data collection and analysis, decision to publish, or preparation of the manuscript.

**Competing interests:** The authors have declared that no competing interests exist.

incorporate water and sanitation improvements into national mental health and public health policies.

## 1. Introduction

The global mental health burden affects nearly 1 billion people, with depression and anxiety being the most common, with 279.6 million (28.8%) and 301.4 million (31.1%) cases, respectively [1]. Over the past 30 years, the prevalence of depression cases has increased by over 1.6 times and anxiety cases have grown by about 1.5 times [1]. This burden is especially pronounced in low- and middle-income countries (LMICs), where over 75% of individuals with mental disorders reside [2]. In these regions, mental health conditions contribute significantly to the overall disease burden, accounting for 16.6% of the overall disease burden in 2019 [1]. Limited resources, insufficient healthcare infrastructure and poor access to mental health services in those countries exacerbate these challenges [3].

In Indonesia, a country with a large and diverse population, approximately 14 million people suffer from depression and anxiety, placing it among the top two countries in the WHO South-East Asia region for the prevalence of these disorders [4]. The impact of mental health issues extends across all aspects of life. Recent studies in Indonesia found that depression is known to reduce work retention by 5.5% [5] and significantly increase the risk of suicide among adults [6]. The financial burden of mental health disorders is significant, with annual treatment costs for depression and anxiety estimated at $85 - $114 per patient, totalling up to $6.244 billion nationally [7]. Such figures highlight the importance of identifying upstream, modifiable determinants of mental health, including environmental and household-level factors.

Among the determinants of mental health, the quality of basic household infrastructure and access to essential services play a critical role in shaping everyday living conditions. National survey data show that Indonesia continues to face notable challenges in achieving universal access to safe water and sanitation. According to SUSENAS 2024 [8], 92.6% of households have access to improved drinking water, but 7.4% still depend on unsafe sources. Recent scoping review evidence also shows that many Indonesian water sources remain contaminated with pollutants, including heavy metals, microplastics, pesticides, endocrine-disrupting chemicals and waterborne pathogens [9]. Toilet ownership is uneven: 89.4% of households have private toilets, 3.1% lack any, with provincial variation ranging from 0.56% to 34.45%, indicating ongoing sanitation issues. Overall, access to improved sanitation reaches 83.6% of households nationally, with significant regional variation from 12.61% to 96.83%. These wide disparities highlight that, despite national progress, significant structural gaps and regional inequality in sanitation access persist.

Understanding the link between water and sanitation-related factors and health is increasingly recognised as a crucial component of public health. Sanitation encompasses access to clean water, proper waste disposal and hygienic living conditions, all of which are vital for both physical and mental well-being [10]. Inadequate

sanitation may lead to adverse health outcomes, including diarrhoea, anaemia and complications such as early pregnancy loss and pre-term birth [11]. Lack of basic sanitation amenities also contributes to psychological distress, anxiety and depression, particularly among women and girls [11]. Evidence from Mozambique, Bangladesh and India shows that inadequate or unsafe sanitation facilities contribute to fear, stress and psychological distress [12–14]. Research from Kenya and Bangladesh further demonstrates that consistent toilet and latrine access improves psychosocial health and happiness [13,15], a pattern reinforced by systematic review evidence linking poor sanitation to anxiety and depression [16]. The connection between poor health and limited access to essential services can also create a vicious cycle, where physical illness exacerbates mental health challenges [17] and poor mental health, in turn, worsens physical conditions [18].

Building on this evidence, psychosocial factors such as life satisfaction (LS) may play a mediating role in the sanitation–mental health pathway. LS is a key component of overall psychological well-being and has been shown to foster resilience and adaptive coping [19,20]. Previous studies have shown that higher LS is positively associated with various aspects of subjective well-being, including psychological functioning, sense of purpose and control, social relationships, and perceived quality of life [20] and resilience [19]. This mechanism aligns with the Transdisciplinary Neighbourhood Health Framework, which posits that environmental conditions influence mental well-being through socio-behavioural mediators such as social support, loneliness and LS, ultimately shaping depressive outcomes [21]. In this study, life satisfaction is conceptualised as a mediating construct because it captures a global cognitive appraisal of one's living conditions [22] and daily circumstances, which is theoretically positioned between environmental exposures and mental health outcomes. From this perspective, improvements in environmental conditions may enhance life satisfaction [22–25], which in turn is associated with a lower risk of depressive symptoms. Applying this lens to sanitation suggests that improvements in environmental conditions may enhance LS [22–24], which in turn mitigates the risk of depressive symptoms.

Despite this growing body of research, there remains a pressing need for comprehensive studies that explore the association between water, sanitation and depression, as current research is limited in scope and geographic coverage. Most studies have been conducted in high-income countries such as the USA [25], Canada [26], and the Netherlands [27]. Such targeted studies limit the generalisability of findings to wider LMIC settings. Furthermore, similar studies conducted in LMICs such as Bangladesh, Brazil, China and Ghana have largely focused on specific populations, such as women, older adults [28–30], or those living in slums [31], which may limit the evidence base needed to inform equitable public health and policy action.

In Indonesia, where disparities in access to clean water and sanitation facilities remain evident [8,32], investigating the potential impact of these issues on mental health is essential. Such research is particularly relevant where elevated mental health needs coincide with environmental constraints, yet empirical evidence linking basic infrastructure to psychological outcomes is scarce. Existing research on sanitation and mental health in Indonesia has primarily focused on sanitation workers [33], emotional disturbances rather than depression [34] and often involved small, non-representative samples of fewer than 100 participants. Beyond access to sanitation, cultural beliefs significantly influence sanitation behaviours. For example, a study in rural Bali found that supernatural beliefs, perceptions of toilets as polluting, and prioritising religious spending over sanitation investments influenced whether households owned toilets [35]. These culturally rooted practices affect not only sanitation behaviour but also feelings of dignity, social acceptance, and perceived purity, psychological dimensions that influence how individuals experience their environment and may impact overall well-being [36]. Given Indonesia's distinctive combination of environmental constraints and culturally shaped sanitation practices, this study provides a contextually grounded examination of how water and sanitation relate to depressive symptoms, with LS as a potential mediator.

This study extends previous work on sanitation and mental health in Indonesia in three key ways. First, unlike prior studies that rely primarily on logistic regression [37] or composite WASH indices [38], we use structural equation modelling (SEM) to quantify both direct and indirect pathways linking sanitation conditions to depressive symptoms. Second, we examine each sanitation component separately rather than as an aggregated index, allowing for a more nuanced

understanding of which aspects of sanitation are most closely associated with depressive symptoms. Third, by incorporating LS as a potential mediator, this study provides new insight into the psychosocial mechanisms that may underlie the sanitation–mental health relationship in the Indonesian context.

## 2. Methods

### 2.1. Data and study population

The data used in this study were obtained from the fifth wave of the Indonesian Family Life Survey (IFLS-5), conducted between September 2014 and March 2015 [39]. The survey is jointly administered by the RAND Corporation (USA), the University of Gadjah Mada and SurveyMETER (both located in Yogyakarta, Indonesia). Initiated in 1998, the IFLS-5 sample represents approximately 83% of the Indonesian population. IFLS-5 provides individual-level data for Indonesians aged 15 and older, including demographics, socio-economic status, sanitation, health and depressive symptoms. The survey used a multistage stratified sampling strategy by province and urban–rural location, randomly selecting household members for detailed interviews [39]. Of the initial 50,000 participants, this study included adults (≥15 years) who completed the Health Conditions (KK), Household Characteristics (KR), Welfare (SW) and Psychological Health (KP) sections, resulting in 31,446 respondents after excluding those with missing data on chronic disease count or depressive symptoms. Access to the data is provided at RAND's website [40], with the research team storing the files in a secure institutional Dropbox folder.

Although IFLS-5 was collected in 2014–2015, it remains one of the few nationally representative datasets in Indonesia that includes detailed household-level measures of water and sanitation conditions alongside validated psychological indicators such as depressive symptoms and LS. These variables are essential for modelling direct and indirect pathways using SEM and are not available together in more recent national surveys. The primary objective of this study is to examine associations, which are conceptually more stable over time, even if national prevalence levels have changed. The use of IFLS-5 is therefore appropriate for this analytical purpose, and the age of the dataset is further acknowledged in the Limitations section.

Missing data were minimal (0.01–6.58%); therefore, we applied listwise deletion, as key variables were categorical and conceptually specific, and imputation could introduce bias [41].

### 2.2. Measurement and variable definitions

#### 2.2.1. Depressive symptoms.
Depressive symptoms were evaluated using the 10-item Center for Epidemiologic Studies Depression Scale (CES-D-10), which assesses the frequency of feelings and behaviours over the past week [42]. Respondents rated each of the ten items on a four-point Likert scale: "rarely or none (≤1 day)", "some days (1–2 days)", "occasionally (3–4 days)", and "most of the time (5–7 days)". Scores for each item ranged from zero to four, with a total score ranging from 0 to 30. A lower total score indicates fewer depressive symptoms. The CES-D-10 has demonstrated strong validity and reliability among the Indonesian population [43]. This is further supported by recent studies using IFLS data, which confirm the psychometric robustness and latent structure of IFLS depressive symptom items [44,45]. Consistent with previous research using the IFLS, respondents were categorised as experiencing depressive symptoms if their CES-D-10 score was 10 or higher [46]. This approach ensures continuity with prior studies and allows for meaningful comparisons across different populations and time periods.

#### 2.2.2. Sanitation conditions.
In the IFLS survey, participants were asked to report their primary sources of drinking water, water used for household activities (such as bathing and washing clothes), their toilet facilities, methods for disposing of liquid waste and methods for disposing of household waste. While the WHO/UNICEF Joint Monitoring Programme (JMP) currently recommends a five-tier service ladder for sanitation facilities, including 'safely managed,' 'basic,' 'limited,' 'unimproved,' and 'no service' [32], not all IFLS survey responses provide sufficient detail to classify households according to these levels. The JMP ladders are primarily designed for international benchmarking and to

capture incremental service quality across countries, building on the traditional classification of facilities as improved or unimproved [32]. However, because the IFLS questionnaire lacks several indicators required for the five-tier system and there are no published studies or policy documents applying the full JMP ladder to the Indonesian context, we were unable to operationalise the five-tier classification in this study.

Following previous studies that have used IFLS sanitation data [37], we dichotomised responses into "safe" versus "unsafe" for drinking water source and waste disposal (liquid and solid) and "improved" versus "unimproved" for house-hold water sources and toilet facilities. For drinking water, we defined it as "safe" if it came from an improved source (pipe, pumped well, spring, rainwater, or bottled mineral water) and/or if the household reported boiling the water before con-sumption and "unsafe" otherwise. This approach captures both the quality of the water source and household treatment practices, while also ensuring comparability with earlier research and a consistent understanding of the quality and safety of water, sanitation and waste management practices across the study population. For toilet facilities, although more gran-ular classifications exist, we followed the dichotomisation approach applied in prior research [47], which demonstrates that reducing toilet facilities into two categories is both analytically valid and widely accepted in public health studies. Table 1 outlines the various responses provided by participants and how these were grouped into categories based on the stan-dard definitions of "improved" or "safe" (coded as 0) and "unimproved" or "unsafe" (coded as 1). S1 Table provides details about the survey questions for each variable.

### 2.2.3. Mediating factor.

**2.2.3.1. Life satisfaction:** Life satisfaction was selected as the mediating variable because it provides a validated cognitive appraisal of overall living conditions and is the only psychosocial construct measured consistently and reliably in IFLS-5. In this study, LS was measured as a mediating variable linking sanitation factors to depressive symptoms. It was assessed using a single-item question from the IFLS, which asked respondents to reflect on their life as a whole and indicate their current level of satisfaction. The question was phrased: "*Reflect on your life as a whole. How satisfied are you with your life right now?*" Responses were recorded on a five-point Likert scale: 1 (Extremely satisfied), 2 (Very

**Table 1. Classifications of sanitation conditions.**

| Variable | Improved/Safe (0) | Unimproved/Unsafe (1) |
|---|---|---|
| **Drinking Water Sources** | Pipe water, well/pump, spring water, rain water, mineral water and those who report boiling the water before drinking. | Well without a pump, river/creek water, pond/fishpond, water collection basin. |
| **Water Source (Household Activities)** | Piped water within household, piped to yard/plot, unpro-tected tube wells, unprotected springs, rainwater collection. | Surface water, unprotected ponds, stored in a tank and other unspecified sources. |
| **Toilet Facilities** | Flush or pour into septic tank. | Flush/pour into pit latrine, shared flush toilet, public flush toilet, or open defecation in places like rivers, gardens, sewers, ponds, cattle pens, oceans, or other unspecified locations. |
| **Liquid Waste Disposal** | Flowing sewers, non-flowing sewers/gutters, permanent pits. | Dumped in yard/garden, ponds, rivers, pud-dles, rice fields, oceans, or other unspecified methods. |
| **Household Waste Disposal** | Rubbish bin collected by waste management officers. | Burned, dumped into rivers, thrown in yard/garden, buried in a hole, forest, sea/lake/beach, rice fields, or other unspecified methods. |

**Source**: IFLS-5 questionnaire items; classifications adapted from previous studies [37,47].

satisfied), 3 (Somewhat satisfied), 4 (Dissatisfied) and 5 (Very dissatisfied). For analysis, the LS was treated as a continuous variable.

While single-item measures may not fully capture the multidimensional nature of quality of life (QoL), they have been shown to demonstrate strong convergent and discriminant validity when compared with multi-item life satisfaction scales and related psychological constructs [48]. In their evaluation, Fonberg & Smith (2019) also reported that single-item LS questions correlate highly with broader subjective well-being indicators, supporting their suitability for use in large population-based studies. This approach has been applied in several peer-reviewed studies using the same IFLS dataset for different analyses [49–51]. These studies demonstrate that the single-item LS indicator is a reliable and pragmatic proxy for overall subjective well-being in large-scale surveys where comprehensive QoL instruments are unavailable.

**2.2.4. Covariates.** In all models, depressive symptoms and LS were adjusted for age and sex, as both variables have been consistently associated with these outcomes in previous studies [52–55]. For sanitation factors, we examined the inclusion of socioeconomic factors (education level, occupation status, residency type and wealth) due to their potential correlation with access to sanitation. However, only the model that included occupation status and residency type achieved acceptable fit indices, while those incorporating wealth and education did not converge or fell below the fit thresholds. Model fit was evaluated using SRMR, RMSEA, CFI and TLI, according to Kline's guidelines [56]. RMSEA values close to or below 0.06 indicate good fit, whereas values above 0.10 suggest poor fit. The Bentler CFI is a goodness-of-fit index; values ≥0.95, in combination with SRMR ≤0.08, reflect acceptable overall model fit. While Tucker-Lewis Index (TLI) values ≥ 0.95 indicate a good fit. Therefore, the final models retained occupation status and residency type alongside age and sex as covariates. Including these demographic controls helps minimise confounding and ensures that the estimated direct and indirect effects of sanitation variables are adjusted for demographic differences. Full goodness-of-fit results from the sensitivity analyses are presented in Supplementary Table S3.

## 2.3. Statistical analysis

Descriptive statistics characterised the sample using frequencies with proportions for categorical data and mean with standard deviation (SD) for continuous variables. Pearson's Chi-squared ($\chi^2$) test and the Kruskal-Wallis test were used to compare nominal/ordinal and continuous variables, respectively, between having depressive and non-depressive symptoms. Structural equation modelling (SEM) was used to examine the relationships between sanitation factors and depressive symptoms. SEM is a widely used analytical method in epidemiological research, particularly well-suited for analysing complex interactions among variables and addressing endogeneity issues [57]. It enables the estimation of direct, indirect and total effects of both exogenous and endogenous variables. Instead of combining all sanitation variables into a single model, we constructed and re-analysed a series of separate models for each sanitation factor. For each factor, we estimated two models: one assessing its direct association with depressive symptoms and another including LS as a mediating variable. This approach enabled us to isolate the unique contribution of each sanitation factor and gain a better understanding of its relationship with both LS and depressive symptoms. The pathways for each model are illustrated in Fig 1, Model A and Model B. We report standardised path coefficients (β) to facilitate comparability across variables and present model fit statistics, including RMSEA, CFI and TLI, with overall goodness-of-fit results provided in S4 Table. Our analysis model fit indices indicated a good overall fit across models, with SRMR values ranging from 0.004 to 0.008, RMSEA values from 0.008 to 0.020, CFI values from 0.972 to 0.994, and TLI values from 0.897 to 0.987. Before performing the SEM analysis, a correlation matrix was examined to assess bivariate associations among sanitation variables, LS and depressive symptoms (S2 Table). SEM analyses were conducted without applying sampling weights. This approach is consistent with prior IFLS-based SEM studies (which typically do not incorporate survey weights) [49,58] and with methodological guidance for analyses focused on modeling relationships and pathways. When the primary objective is to examine theoretical associations rather than to produce precise population-level descriptive estimates, sampling weights may be unnecessary and can reduce estimator efficiency [59]. All analyses were performed using Stata 17.

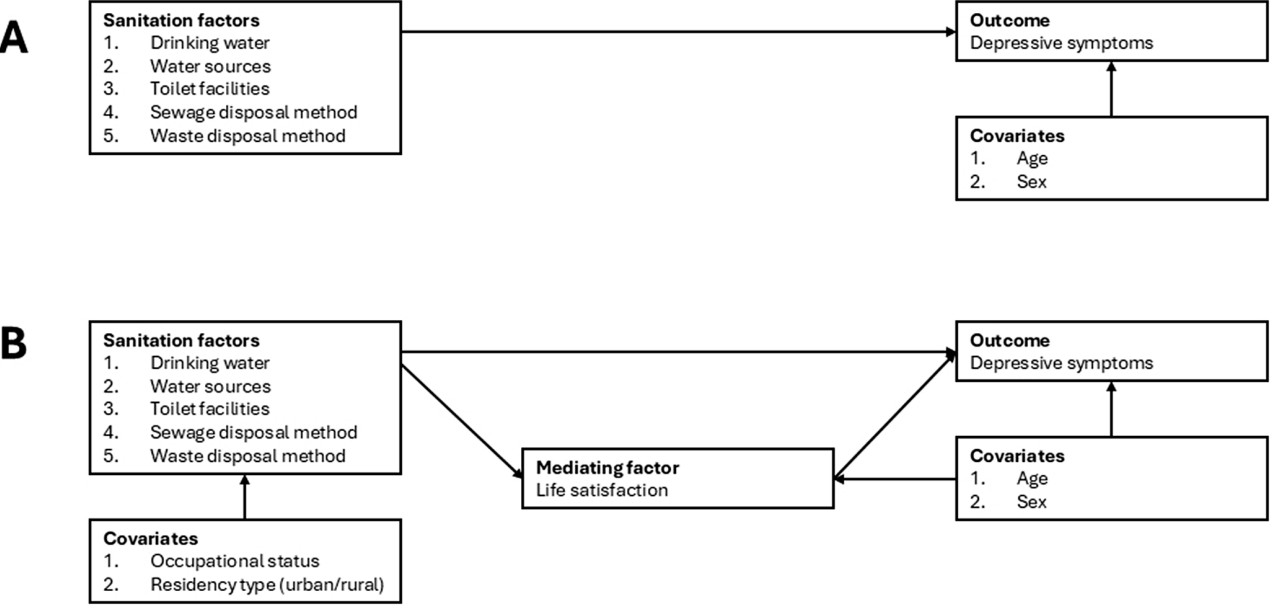

**Fig 1. SEM pathways: (A)** Direct effect of sanitation on depressive symptoms; **(B)** Indirect effect via LS.

## 2.4. Ethics statement

The IFLS has been approved by ethics review boards at both the RAND Corporation and Gadjah Mada University in Indonesia and written informed consent was obtained from all respondents before data collection. The ethical review boards of RAND's Human Subjects Protection Committee (s0064-06–01-CR01) also approved the IFLS research. As the study utilised publicly available IFLS datasets containing no personal identification, no further ethical approval was required. The data source maintained the confidentiality and anonymity of participants' names, addresses, locations and neighbourhoods, in accordance with RAND's human subjects' protection rules. Respondents were assured that their answers would remain confidential and that their identities would be linked only to an anonymous code. The use of the data set for this study has also been approved by the University of Manchester, the Faculty of Nursing Universitas Indonesia and the National Research and Innovation Agency of Indonesia (BRIN).

## 3. Results

### 3.1. Sample characteristics and bivariate analysis

Table 2 presents the descriptive characteristics stratified by depressive symptoms status. In total, 53% of the sample consisted of women, with an average age of 37.32 years (±14.93 years), based on unweighted data. The final analytic sample included respondents aged 15–101 years, with no age groups excluded except for cases removed due to missing data. Most of the sample resides in urban areas (59%), has completed senior high school (55%) and is employed (67%). Regarding access to basic amenities, the majority reported using safe drinking water (98%), improved water sources (95%), improved toilet facilities (74%) and safe liquid waste disposal (72%). However, only 35% of the sample have access to improved waste disposal. Details on combinations of water and sanitation issues, as well as the cumulative number of issues reported by participants, are provided in Supplementary S5 and S6 Tables.

**Table 2. Descriptive statistics on the individual characteristics.**

| Variables | Total (n = 31,446) | | No depressive symptoms (n = 24,134) | | With depressive symptoms (n = 7,312) | | P value* | Missing data |
|---|---|---|---|---|---|---|---|---|
| | N | %/SD | n/mean | %/SD | n/mean | %/SD | | N (%) |
| Safe drinking water | 31,024 | 98.65 | 23,846 | 98.80 | 7,178 | 98.17 | <0.001 | 44 (0.13) |
| Improved water source | 29,892 | 95.06 | 23,013 | 95.35 | 6,879 | 94,08 | <0.001 | 44 (0.13) |
| Improved toilet facility | 23,448 | 74.56 | 18,236 | 75.56 | 5,212 | 71.28 | <0.001 | 44 (0.13) |
| Safe liquid waste disposal | 22,643 | 72.00 | 17,526 | 72.62 | 5,117 | 69.98 | <0.001 | 44 (0.13) |
| Safe waste disposal | 11,115 | 35.35 | 8,634 | 35.77 | 2,481 | 33.93 | 0.015 | 44 (0.13) |
| Urban residence | 18,558 | 59,02 | 14,207 | 58.87 | 4,351 | 59.50 | 0.331 | – |
| Age (years), mean (SD) | 37.32 | 14.93 | 38.16 | 15.03 | 34.56 | 14.24 | <0.001 | 8 (0.02) |
| Sex | | | | | | | 0.005 | – |
| Male | 14,700 | 46.76 | 11,403 | 47.25 | 3,297 | 45.09 | | |
| Female | 16,738 | 53.24 | 12,725 | 52.73 | 4,013 | 54.88 | | |
| Education | | | | | | | <0.001 | 1,222 (3.88) |
| Primary school and lower | 9,291 | 30.74 | 7,099 | 29.41 | 2,192 | 29.98 | | |
| Senior high school and higher | 16,578 | 54.85 | 12,612 | 52.26 | 3,966 | 54.24 | | |
| College | 4,355 | 14,41 | 3,433 | 14.22 | 922 | 12.61 | | |
| Occupation status | | | | | | | 0.055 | 2 (0.01) |
| Employed | 21,031 | 66,88 | 16,220 | 67.21 | 4,811 | 65.80 | | |
| Unemployed | 10,413 | 33,12 | 7,913 | 32.79 | 2,500 | 34.19 | | |
| Economic status | | | | | | | <0.001 | 2,071 (6.58) |
| Quintile 5 | 5,910 | 20.12 | 4,608 | 19.09 | 1,302 | 17.81 | | |
| Quintile 4 | 5,931 | 20.19 | 4,600 | 19.06 | 1,331 | 18.20 | | |
| Quintile 3 | 5,911 | 20.12 | 4,503 | 18.66 | 1,408 | 19.26 | | |
| Quintile 2 | 5,863 | 19.96 | 4,502 | 18.65 | 1,361 | 18.61 | | |
| Quintile 1 | 5,760 | 19.61 | 4,300 | 17.82 | 1,460 | 19.97 | | |
| Life satisfaction | | | | | | | <0.001 | 4 (0.01) |
| Extremely satisfied | 1,340 | 4.26 | 1,043 | 4.32 | 297 | 4.06 | | |
| Very satisfied | 12,364 | 39.32 | 10,026 | 41.54 | 2,338 | 31.97 | | |
| Somewhat satisfied | 13,346 | 42.44 | 10,235 | 42.41 | 3,111 | 42.55 | | |
| Dissatisfied | 3,889 | 12.37 | 2,558 | 10.60 | 1,331 | 18.20 | | |
| Very dissatisfied | 504 | 1.60 | 271 | 1.12 | 233 | 3.19 | | |

**Note:** * Bivariate analysis was performed using Chi-square for categorical variables and Kruskal-Wallis for continuous variables.

**Source**: IFLS-5 (2014–2015), authors' calculations.

The bivariate analysis (last column of Table 2) reveals that individuals with depressive symptoms are less likely to have access to safe drinking water (p ≤ 0.001), improved water sources (p = 0.023), improved toilet facilities (p ≤ 0.001) and improved waste disposal methods (p = 0.004). Furthermore, there is a significant gender disparity in depressive symptoms rates, with a higher proportion of females affected compared to males (p = 0.005). Age also plays a crucial role, as younger individuals are more prone to depressive symptoms (p ≤ 0.001). Additionally, individuals with depressive symptoms are more likely to have lower educational attainment, with 54% having completed senior high school compared to only 13% being college graduates. Economic disparities are evident, with those in lower economic quintiles more likely to suffer from depressive symptoms (20% in quintile 1 vs 18% in quintile 5). Lastly, the prevalence of depressive symptoms was

substantially higher among those who reported being dissatisfied or very dissatisfied with life, supporting the notion that depressive symptoms and LS follow an inverse and potentially U-shaped relationship.

### 3.2. SEM analysis

Detailed results from the SEM analysis, including direct and indirect effects, are presented in Table 3. In the first model (Fig 1, Model A), we found that all sanitation variables, drinking water (β = 0.022, p < 0.001), water source (β = 0.024, p < 0.001), toilet facilities (β = 0.039, p < 0.001), sewage disposal method (β = 0.027, p < 0.001) and waste disposal method (β = 0.027, p < 0.001) were significantly associated with depressive symptoms. These findings suggest that inadequate access to safe water, improved sanitation access and appropriate sewage or waste management are important factors associated with depressive symptoms.

In the second SEM model (Fig 1, Model B), significant direct effects of unimproved sanitation on depressive symptoms were observed for drinking water (β = 0.021, p < 0.001), water source (β = 0.026, p < 0.001), toilet facilities (β = 0.031, p < 0.001), sewage disposal (β = 0.022, p < 0.001) and waste disposal method (β = 0.015, p < 0.05). These results are consistent with the direct pathways identified in Model A, reinforcing that these five sanitation factors exert a direct influence on depressive symptomseven when LS is included as a mediator.

Regarding the relationship between sanitation factors and LS, unimproved sanitation was also significantly associated with lower LS, especially for drinking water (β = 0.013, p < 0.05), water source (β = 0.054, p < 0.001), toilet facilities (β = 0.068, p < 0.001) sewage disposal (β = 0.034, p < 0.001) and waste disposal methods (β = 0.046, p < 0.001), suggesting a possible indirect pathway. Finally, LS was associated with depressive symptoms across all sanitation factors (β ≈ 0.124–0.126, p < 0.001), confirming its mediating role in the sanitation–depressive symptoms relationship. These results suggest that the effect of unimproved sanitation on depressive symptoms is partly mediated through reduced LS, though substantial direct effects remain for several sanitation dimensions. Supplementary analyses, including logistic regression models (S7 Table), supported these findings with consistent associations and a sub-sample SEM for the lowest wealth quintile (S8 Table) showed similar patterns. Some effects became non-significant, likely due to a smaller sample size and less variability in the lower-income group.

## 4. Discussion

This study is the first to investigate the relationship between sanitation and depression in Indonesia, employing SEM analysis with LS as a mediator. The findings contribute new insights into the pathways linking sanitation, LS and mental health. The following discussion integrates these results with recent empirical literature, including reviewer-suggested studies and highlights implications for theory, policy and future research.

**Table 3. Standardised Coefficients for Direct Effects of Sanitation on Depressive Symptoms (Model A) and Indirect Effects via LS as a Mediator (Model B), SEM Analysis.**

| Sanitation Variables | Direct – Model A | Indirect – Model B | | |
|---|---|---|---|---|
| | Depressive symptoms | Sanitation → Depressive symptoms | Sanitation → LS | LS → Depressive symptoms |
| **Drinking water** | 0.022 (0.011–0.033)* | 0.021 (0.010–0.032)* | 0.013 (0.001–0.023)+ | 0.125 (0.114–0.136)* |
| **Water source** | 0.024 (0.013–0.035)* | 0.021 (0.010–0.032)* | 0.023 (0.012–0.034)* | 0.125 (0.114–0.136)* |
| **Toilet facilities** | 0.039 (0.028–0.050)* | 0.031 (0.020–0.042)* | 0.068 (0.057–0.079)* | 0.124 (0.113–0.135)* |
| **Sewage disposal method** | 0.027 (0.016–0.038)* | 0.022 (0.011–0.033)* | 0.034 (0.023–0.045)* | 0.125 (0.119–0.136)* |
| **Waste disposal method** | 0.021 (0.010–0.032)* | 0.015 (0.004–0.026)+ | 0.046 (0.035–0.057)* | 0.125 (0.114–0.136)* |

**Note**: Standardised coefficient; +p < 0.05, #< 0.005, *p < 0.001. Sanitation factors were adjusted with employment status and residency type, while LS and depressive symptoms were adjusted with age and sex.

**Source**: Authors' analysis of IFLS-5 data.

## 4.1. Sanitation and depressive symptoms

The SEM analysis indicates that unsafe drinking water, unimproved water sources, unimproved toilets and unsafe waste disposal were directly associated with higher odds for depressive symptoms. Several sanitation factors were also associated with lower LS, which in turn was strongly linked to depressive symptoms, confirming its role as a mediator in the sanitation–mental health pathway. This study strengthens the existing literature that links poor sanitation conditions to increased levels of depression [25,27,29–31,37,60–63], highlighting the importance of improving sanitation to enhance mental health outcomes.

We identified a significant association between unimproved drinking water quality and depressive symptoms, consistent with studies which demonstrated that consumption of water contaminated with heavy metals such as copper, cadmium [30], lead, nitrate and arsenic [25] was associated with higher rates of depression. Reflecting similar findings in Indonesia, a study showed that relying on unimproved water sources and drinking unboiled water considerably raised the risk of depression [37]. Beyond drinking water quality, we also found that the use of unimproved water sources for daily activities was significantly associated with higher odds of depression. This finding aligns with studies from Canada [26] and South Africa [62], which reported that individuals with piped household water sources had lower odds of depression. Further evidence from the China Family Panel Studies showed that access to clean water was associated with lower levels of depression [60].

Unsafe drinking water may affect mental health through multiple interconnected pathways. First, unsafe water compromises physical health by increasing the risk of diarrhoea, acute respiratory infections and undernutrition [64], which may indirectly contribute to depression through reduced energy, chronic illness and feelings of hopelessness. Second, inadequate or unsafe water access acts as a persistent external stressor, generating fear of disease, household conflict and even trauma associated with water-related illness [65]. Beyond physical health, limited access to safe water and adequate sanitation can heighten psychosocial stress, including worry, shame and reduced privacy, which may increase anxiety and, over time, contribute to depressive symptoms [66]. For women in particular, water insecurity has been linked to increased fear of harassment or physical violence when collecting water, which exacerbates anxiety and psychological distress [16].

Furthermore, this study identified a significant direct association between the use of unimproved toilet facilities and depressive symptoms, which aligns with studies conducted globally. A WHO SAGE wave 2 study reported that individuals using unimproved toilet facilities were more likely to report major depressive episodes [29]. Additionally, a study in China indicated that deficiencies in village infrastructure, including inadequate drinking water and toilet facilities, were positively associated with higher depression rates [61]. Studies from Kenya, Bangladesh and Dhaka further support these findings, showing that regular access to better-quality toilets and latrines has a significant positive impact on mental health and well-being [13,15,31].

A plausible mechanism linking unimproved sanitation to poor mental well-being is outlined in Sclar's conceptual model [67]. Lack of privacy and safety during open defecation or using inadequate facilities can cause anxiety, shame and loss of dignity. This issue is even more severe among women and girls who face risks during urination, defecation and menstruation [25]. Fear and anxiety often accompany perceived risks to safety, while actual violations, such as poorly maintained facilities or the need to practice open defecation, exacerbate feelings of shame and psychological stress, ultimately contributing to depression [16]. These experiences are further shaped by factors such as gender identity, physical ability, life stage, residency status and socioeconomic status [67].

Further analysis revealed a significant pathway between the use of unimproved sewage and waste disposal methods and depressive symptoms. Methods such as burning waste, dumping it into rivers, disposing of it in gardens, or burying it in holes can lead to significant pollution near residential areas. A study from South Africa discovered that residing within 5 km of a waste site was significantly associated with depression [63]. Similarly, a Dutch study found that higher levels of odour annoyance from sources such as sewerage and others were linked to increased psychological distress and a higher

likelihood of clinical depression [27]. Improper waste disposal can contribute to depression through multiple pathways. Open burning of mixed household waste releases harmful air pollutants, such as particulate matter (PM) [68]. Exposure to such air pollutants was known to be significantly associated with increased severity of depression among asthma patients [69] and even higher risk of suicide [70]. Beyond air pollutants, waste left in open dumps releases toxic gases and foul odours, which may heighten psychological stress, reduce LS and contribute to emotional distress, thereby increasing vulnerability to depression [71,72].

Overall, our findings support existing evidence from Bangladesh [13], Ghana [29] and similar contexts [15,31], indicating that Individuals using poor-quality toilet facilities tend to report higher depressive symptoms. By analysing sanitation components individually rather than treating water and sanitation variables as a single index, our study demonstrates that toilet facility quality itself is a significant and correlate of depressive symptoms in Indonesia. This component-level approach provides a more detailed understanding of which sanitation deficits carry the most significant psychological burden, offering more precise and actionable targets for national WASH and mental health interventions.

### 4.2. Life satisfaction as a mediator

The current study showed that LS partially mediates the relationship between sanitation factors and depressive symptoms. Specifically, better sanitation was associated with higher LS, which in turn was linked to lower depressive symptoms. Our finding that sanitation is linked to LS aligns with other LMIC studies [22–24]. For example, a recent study in Pakistan likewise found that access to piped and bottled water was positively associated with subjective well-being in both men and women and that improved sanitation was strongly linked to higher LS, especially among wome n[22]. Similarly, an analysis in Indonesia showed households with private improved latrines had the highest LS, while those sharing facilities or practising open defecation reported lower LS [23]. Complementing these findings, a Chilean study revealed that residential air pollution was strongly and consistently negatively associated with LS, particularly in areas where non-emitting households are exposed to pollution generated by others [24].

Plausible mechanisms help explain how sanitation influences LS. Improved sanitation not only enhances physical conditions but also contributes to psychological and social well-being [73]. Access to clean and private facilities enhances comfort and self-esteem, while also reducing psychological distress. These improvements can indirectly boost LS by promoting better health, reducing healthcare costs and saving time that would otherwise be spent coping with inadequate facilities [73]. Additionally, a qualitative study highlights that sanitation shapes LS by creating a sense of safety and control over one's environment, alleviating shame and stress and ensuring privacy, all of which foster a more secure, dignified and emotionally supportive living context [12].

In this study, we found that LS was strongly associated with depressive symptoms across all sanitation factors, consistent with evidence from other settings. In Kampala, Uganda, lower LS were associated with higher stress levels and frequent depressive symptoms [74]. Similarly, a study in Korea [75] and Portugal [76] found that participants with higher LS had a lower risk of depression. These findings are supported by research suggesting plausible mechanisms through which LS influences depression: higher LS appears to protect against depressive symptoms by enhancing resilience, positive coping and social support, while reducing vulnerability to negative emotional states [77,78]. LS may also indirectly influence depression by supporting daily functioning and overall well-being [79].

### 4.3. Theoretical and practical implications

The mediation model examined in this study aligns with recent empirical frameworks highlighting the role of sanitation factors in shaping mental health outcomes [65,67,70]. While prior research has often focused on the direct impact of sanitation on physical health or depression in isolation, few studies, particularly in LMICs, have empirically examined psychosocial pathways through which sanitation may relate to depressive symptoms. By testing LS as a mediator and modelling each sanitation component separately using SEM, this study advances the literature by demonstrating that environmental

living conditions may influence mental well-being through broader subjective evaluations of life, offering a more nuanced theoretical perspective than traditional regression-based approaches.

From a practical perspective, the findings remain relevant to Indonesia's current WASH landscape. National SUSE-NAS data show persistent disparities in safely managed sanitation, especially outside major urban areas, despite ongoing programmes such as the Residential Sanitation Development Acceleration Program (PPSP, *Program Percepatan Pembangunan Sanitasi Permukiman*) and Community-Based Total Sanitation (STBM, *Sanitasi Total Berbasis Masyarakat*). Evidence from Li et al. indicates that better village infrastructure and community resources are associated with fewer depressive symptoms, highlighting the mental health benefits of such investments [61]. As Indonesia works toward Sustainable Development Goal 6 and universal access to improved water and sanitation services, integrating these initiatives with mental health promotion could yield dual benefits, consistent with the WHO, which further estimated that every US$1 invested in sanitation yields a return of US$5.50 through reduced healthcare costs, higher productivity and fewer premature deaths [10]. Integrating water and sanitation programmes into national mental health and development agendas could therefore generate both health and economic gains, particularly in underserved rural settings.

### 4.4. Strengths, limitations and directions for future research

The strength of this study lies in its use of a large, nationally representative sample from Indonesia, enhancing the generalisability of its findings to similar LMIC contexts. By examining a range of sanitation factors individually, this study provides a detailed assessment of their direct associations with depressive symptoms and their indirect effects mediated through LS. The use of SEM further enables the simultaneous estimation of these pathways, providing insights into the mechanisms that link environmental conditions to mental health outcomes.

However, this study also has notable limitations. The data were collected in 2014–2015 and changes in sanitation infrastructure and mental health awareness since then may affect the current relevance of the findings. Additionally, the cross-sectional nature of the data limits the ability to make causal inferences. Third, the IFLS only covers 83% of the population and excludes eastern regions of Indonesia, where sanitation conditions are likely to be more challenging. Although missing data were minimal (0.01–6.58%), the use of listwise deletion may have slightly reduced the analytic sample. Furthermore, LS was measured using a single-item indicator, which may not fully capture the multidimensional nature of subjective well-being. These limitations may affect the comprehensiveness and accuracy of the findings, potentially skew the results and limit the generalisability of the study's conclusions.

Future research should build on these findings by using more recent and longitudinal data to capture changes in sanitation infrastructure and mental health trends, enabling stronger causal inference. Expanding coverage to eastern regions of Indonesia and other underrepresented populations would enhance generalisability. Additionally, exploring other determinants of depressive symptoms, such as social support, economic status and environmental stressors, could provide a more comprehensive understanding of how sanitation affects mental health.

### 5. Conclusion

Our study highlights the multifaceted relationship between water, sanitation and mental health in Indonesia, revealing that inadequate access to safe drinking water, improved sanitation facilities and proper waste management are significantly associated with depressive symptoms. Even after accounting for LS as a mediator, these environmental factors maintained strong direct effects on depressive symptoms, underscoring their independent influence. The results also demonstrate that unimproved sanitation contributes to lower LS, which in turn exacerbates depressive symptoms, suggesting that the mental health burden linked to poor sanitation operates through both direct and psychosocial pathways. Our research suggests that expanding access to safe drinking water and sanitation is a cost-effective way to improve population health, with evidence showing mental health benefits and high economic returns on investment. Integrating these initiatives into national policies, especially in underserved rural areas, could lead to substantial health and financial gains.

## Supporting information

**S1 Table. Complete Sanitation and Water Safety Classification Table.**
(DOCX)

**S2 Table. Correlation Matrix.**
(DOCX)

**S3 Table. Overall goodness-of-fit statistics for the sensitivity analyses.**
(DOCX)

**S4 Table. Overall goodness-of-fit statistics of each model.**
(DOCX)

**S5 Table. Combinations of Water and Sanitation Issues.**
(DOCX)

**S6 Table. Cumulative Sanitation Issues and Depression.**
(DOCX)

**S7 Table. Robustness Check Using Logistic Regression Models.**
(DOCX)

**S8 Table. Structural Equation Model Results Among the Poorest 20th Percentile.**
(DOCX)

## Acknowledgments

This research was based on the IFLS-5 conducted by RAND (http://www.rand.org/labor/FLS/IFLS.html). The authors extend their gratitude to RAND for providing access to the survey data and to the study participants for their valuable contributions.

## Author contributions

**Conceptualization:** Mashita Fajri, Asri Maharani.

**Data curation:** Mashita Fajri.

**Formal analysis:** Mashita Fajri, Asri Maharani.

**Funding acquisition:** Herni Susanti, Helen Brooks, Penny Bee.

**Investigation:** Mashita Fajri, Sri Idaiani, Jonathan Gibson, Asri Maharani.

**Supervision:** Sri Idaiani, Herni Susanti.

**Writing – original draft:** Mashita Fajri, Asri Maharani.

**Writing – review & editing:** Sri Idaiani, Amy Blakemore, Jonathan Gibson, Herni Susanti, Helen Brooks, Penny Bee.

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
