## [Decision Letter · Decision Letter 0]

17 Nov 2025

Dear Dr. Maharani,

Thank you for submitting your manuscript to PLOS ONE. After careful consideration, we feel that it has merit but does not fully meet PLOS ONE’s publication criteria as it currently stands. Therefore, we invite you to submit a revised version of the manuscript that addresses the points raised during the review process.

We look forward to receiving your revised manuscript.

Kind regards,

Alison Parker

Academic Editor

PLOS ONE

**Journal Requirements:**

“This research was funded by the NIHR [Global Health Research for Sustainable Care for anxiety and depression in Indonesia (Award ID NIHR134638)] using UK international development funding from the UK Government to support global health research. The views expressed in this publication are those of the author(s) and not necessarily those of the NIHR or the UK government.Please state what role the funders took in the study.”

If the funders had no role, please state: "The funders had no role in study design, data collection and analysis, decision to publish, or preparation of the manuscript."

3. Please expand the acronym “NIHR” (as indicated in your financial disclosure) so that it states the name of your funders in full.

4. Please note that funding information should not appear in any section or other areas of your manuscript. We will only publish funding information present in the Funding Statement section of the online submission form. Please remove any funding-related text from the manuscript.

**Additional Editor Comments:**

Three reviewers have provided detailed and constructive comments which will improve your manuscript.

Reviewers' comments:

Reviewer's Responses to Questions

**Comments to the Author**

1. Is the manuscript technically sound, and do the data support the conclusions?

Reviewer #1: Yes

Reviewer #2: Partly

Reviewer #3: Yes

2. Has the statistical analysis been performed appropriately and rigorously?

Reviewer #1: No

Reviewer #2: I Don't Know

Reviewer #3: Yes

3. Have the authors made all data underlying the findings in their manuscript fully available?

Reviewer #1: Yes

Reviewer #2: Yes

Reviewer #3: Yes

4. Is the manuscript presented in an intelligible fashion and written in standard English?

Reviewer #1: Yes

Reviewer #2: Yes

Reviewer #3: Yes

Reviewer #1: Review

Water, Sanitation, and Depression in Indonesia: The Mediating Role of Life Satisfaction

Abstract

Line 31: Please add the duration of observation. How long was the questionnaire data collected?

Line 36: Correct the numerical error: (β=0.54) → (β=0.054).

Introduction

Lines 63–69: The information presented is relevant to the title. However, paragraphs 2 and 3 appear to lack coherence. The connection between the discussion of depression in Indonesia (paragraph 2) and issues of water and sanitation (paragraph 3) is somewhat abrupt. It would be helpful to include data sources related to population size, water quality vulnerability, and sanitation health in Indonesia.

Lines 80–89: Is the explanation of the mediating role of life satisfaction (LS) in the model theoretically robust enough? The authors should clarify why LS was selected as the primary mediator instead of other social factors, such as social support or environmental stress.

Method

Table 1: For liquid waste disposal, the “safe” category lists discharge into rivers, while the “unsafe” category also includes discharge into rivers. The authors should clarify the distinction between these two classifications.

Lines 115–134: The explanation of the IFLS dataset is comprehensive; however, the access time (“November 2024”) appears inconsistent with the 2014–2015 data. This should be corrected to avoid the impression that the data were accessed after the analysis had been conducted.

Ensure consistent use of “Model 1” and “Model 2” or “Model A” and “Model B.” In the Methods section, the text refers to Models 1 and 2, whereas in the Analysis section, it switches to Models A and B.

Result and Discussion

The discussion remains overly descriptive and does not sufficiently emphasise the study’s novelty compared with previous research. The authors could add a concluding paragraph to Section 4.1 highlighting the contribution of this study relative to research conducted in Bangladesh or Ghana.

Spelling and Sentence Structure

1. Lines 231-232: Please use the present tense in the entire result sentence, e.g. "Table 2 presents..."

2. Lines 381-383: The sentence "WHO further estimated that every US$1 invested...", please change to "The WHO further estimates..." in order to parallel the academic style.

3. Use the term “life satisfaction (LS)” consistently throughout the manuscript. Some sections spell it out in full without the abbreviation.

4. Add a source note to each table (e.g., Data source: IFLS-5, authors’ calculation).

5. In British style, the use of a comma before and is not required unless necessary to prevent ambiguity.

6. Line 274: “Standardized coefficient” � “Standardised coefficient”

7. Line 384: “sanitation programs…” � “sanitation programmes…”

8. Please review and ensure consistency with British English conventions across the manuscript.

9. Improve sentence structure for clarity and conciseness, avoiding redundancy.

References

Please use the most recent references to support your research. References to numbers [25], [32], [44], [58], and [59] used over the last 10 years.

In example:

Sakti, B., Wibawa, S., Maharani, A. T., Andhikaputra, G., Savira, M., Putri, A., Iswara, A. P., Sapkota, A., Sharma, A., Syafei, A. D., & Wang, Y. (2023). Effects of Ambient Temperature , Relative Humidity , and Precipitation on Diarrhea Incidence in Surabaya. International Journal of Environmental Research and Public Health, 20, 2313.

Wulandari, R., Iswara, A. P., Qadafi, M., Prayogo, W., Astuti, R. D. P., Utami, R. R., Jayanti, M., Awfa, D., Suryawan, I. W. K., Fitria, L., & Andhikaputra, G. (2024). Water pollution and sanitation in Indonesia: a review on water quality, health and environmental impacts, management, and future challenges. Environmental Science and Pollution Research, 0123456789. https://doi.org/10.1007/s11356-024-35567-x

Reviewer #2: The manuscript explores an important and underexamined topic linking water and sanitation conditions to depressive symptoms in Indonesia using IFLS-5 data. The study is well-motivated and methodologically relevant, but several key issues need revision I have highlight them in the file.

Reviewer #3: This manuscript addresses an important and timely topic by examining the relationships between water access, sanitation, life satisfaction, and depression in Indonesia using IFLS-5 data. The study is methodologically sound, the use of SEM is appropriate, and the paper is clearly written. The topic is highly relevant for Indonesia and contributes to the broader literature on environmental and psychosocial determinants of mental health.

However, several areas require clarification and strengthening. The Introduction needs better coherence (especially when discussing cultural influences) and more explicit novelty. The Methods section should justify the use of 2014–2015 data and clarify key details such as age distribution, missing data handling, and the rationale for simplifying JMP sanitation indicators. Several results also require clearer explanation. Finally, the Theoretical and Practical Implications section should be expanded to better reflect current conditions in Indonesia and highlight the study’s contribution. Overall, this is a valuable paper that can be improved with moderate revisions.

**Do you want your identity to be public for this peer review?** For information about this choice, including consent withdrawal, please see our Privacy Policy

Reviewer #1: No

Reviewer #2: No

Reviewer #3: **Yes:** Satriani

---

## [Author Response · Author response to Decision Letter 1]

22 Dec 2025

Comments from Editor #1

https://journals.plos.org/plosone/s/file?id=wjVg/PLOSOne_formatting_sample_main_body.pdf [journals.plos.org] and

https://journals.plos.org/plosone/s/file?id=ba62/PLOSOne_formatting_sample_title_authors_affiliations.pdf [track.editorialmanager.com]

Authors’ response

Thank you for the reminder. We have revised the manuscript and accompanying files to ensure full compliance with PLOS ONE’s formatting and file-naming requirements.

Comments from Editor #2

Thank you for stating the following financial disclosure:

“This research was funded by the NIHR [Global Health Research for Sustainable Care for anxiety and depression in Indonesia (Award ID NIHR134638)] using UK international development funding from the UK Government to support global health research. The views expressed in this publication are those of the author(s) and not necessarily those of the NIHR or the UK government. Please state what role the funders took in the study.”

If the funders had no role, please state: "The funders had no role in study design, data collection and analysis, decision to publish, or preparation of the manuscript."

If this statement is not correct you must amend it as needed. Please include this amended Role of Funder statement in your cover letter; we will change the online submission form on your behalf.

Authors’ response

Thank you for the clarification. We confirm that the funders had no role in the study design, data collection and analysis, the decision to publish, or the preparation of the manuscript. We have included the required statement in the cover letter as requested. The funder statement thus revised as: “This research was funded by the National Institute for Health and Care Research (NIHR) [Global Health Research for Sustainable Care for anxiety and depression in Indonesia (Award ID NIHR134638)] using UK international development funding from the UK Government to support global health research. The views expressed in this publication are those of the author(s) and not necessarily those of the NIHR or the UK government. The funders had no role in study design, data collection and analysis, decision to publish, or preparation of the manuscript.” We have also ensured that this updated funding and role-of-the-funder statement is reflected in the online submission system.

Comments from Editor #3

Please expand the acronym “NIHR” (as indicated in your financial disclosure) so that it states the name of your funders in full.

Authors’ response

We have expanded the acronym NIHR to its complete form, National Institute for Health and Care Research, in the revised funding statement and cover letter as requested.

Comments from Editor #4

Please note that funding information should not appear in any section or other areas of your manuscript. We will only publish funding information present in the Funding Statement section of the online submission form. Please remove any funding-related text from the manuscript.

Authors’ response

We have removed all funding-related text from the manuscript, in accordance with the journal’s guidelines.

Comments from Editor #5

Your ethics statement should only appear in the Methods section of your manuscript. If your ethics statement is written in any section besides the Methods, please move it to the Methods section and delete it from any other section. Please ensure that your ethics statement is included in your manuscript, as the ethics statement entered into the online submission form will not be published alongside your manuscript.

Authors’ response

Thank you for the guidance. We have moved the ethics statement to the Methods section as requested and removed it from all other sections.

Comments from Editor #6

Authors’ response

Thank you for the suggestion. We have reviewed all publications recommended by the reviewers and have cited those that are relevant to the scope and analysis of our study, while excluding works that did not align with the manuscript’s focus.

The relevant reviewer-recommended references have been incorporated and are provided below:

Wulandari R, Iswara AP, Qadafi M, Prayogo W, Astuti RDP, Utami RR, et al. Water pollution and sanitation in Indonesia: a review on water quality, health and environmental impacts, management, and future challenges. Environmental Science and Pollution Research. 2024;31(58):65967-92. doi: 10.1007/s11356-024-35567-x.

Fakhrunnisak D, Patria B. The positive effects of parents' education level on children's mental health in Indonesia: a result of longitudinal survey. BMC Public Health. 2022;22(1):949. doi: 10.1186/s12889-022-13380-w.

Marvianto R, Kurniawan D, Husein M. Unveiling the Characteristics of Depression Symptoms in Indonesia Population: Lesson Learned from the 5th Wave of the Indonesian Family Life Survey (IFLS-5). SSRN. 2025. doi: 10.2139/ssrn.5365151.

Comments from 1st Reviewer #1

Abstract: Line 31: Please add the duration of observation. How long was the questionnaire data collected?

Authors’ response

Thank you for this helpful suggestion. We have now added the IFLS-5 data collection period (September 2014 to March 2015) to both the Abstract and Methods sections.

Comments from 1st Reviewer #2

Abstract: Line 36: Correct the numerical error: (β=0.54) → (β=0.054).

Authors’ response

Thank you for pointing out this error. We have corrected the coefficient to β = 0.054 in the Abstract.

Comments from 1st Reviewer #3

Introduction: Lines 63–69: The information presented is relevant to the title. However, paragraphs 2 and 3 appear to lack coherence. The connection between the discussion of depression in Indonesia (paragraph 2) and issues of water and sanitation (paragraph 3) is somewhat abrupt. It would be helpful to include data sources related to population size, water quality vulnerability, and sanitation health in Indonesia.

Authors’ response

Thank you for this helpful comment. The transition from the discussion of depressive symptoms to the subsequent section on water and sanitation has been strengthened in the revised manuscript by explicitly framing sanitation as a determinant of mental health and by incorporating nationally representative data to contextualise Indonesia’s current water and sanitation conditions. These include updated SUSENAS 2024 estimates on access to improved drinking water, toilet ownership, and improved sanitation.

Comments from 1st Reviewer #4

Introduction: Lines 80–89: Is the explanation of the mediating role of life satisfaction (LS) in the model theoretically robust enough? The authors should clarify why LS was selected as the primary mediator instead of other social factors, such as social support or environmental stress.

Authors’ response

Thank you for this valuable comment. We have clarified the theoretical justification for using life satisfaction (LS) as the primary mediator in the introduction. LS is a widely used cognitive indicator of subjective well-being, encompassing multiple dimensions such as psychological functioning, sense of purpose and control, social relationships, and perceived quality of life, and has a well-established conceptual link to both environmental conditions and mental health outcomes. Theoretically, LS reflects an individual’s overall appraisal of their living environment, resources, and daily circumstances, making it a suitable intermediary between household sanitation conditions and depressive symptoms.

Regarding alternative mediators, IFLS-5 does not include a validated measure of social support (e.g., MSPSS or ISEL). The variables available in IFLS, such as social participation and neighbourhood trust, capture dimensions of social capital rather than interpersonal or emotional support and therefore do not align conceptually with the mediating role proposed in our pathway. While some prior IFLS-based studies have incorporated external environmental exposure data (e.g., satellite-derived PM₂․₅ or wildfire smoke estimates), IFLS itself does not collect validated measures of perceived environmental stress, neighbourhood disorder, or climate-related stress that could serve as psychosocial mediators within an SEM pathway. These clarifications have been added to the Methods section.

Comments from 1st Reviewer #5

Method: Table 1: For liquid waste disposal, the “safe” category lists discharge into rivers, while the “unsafe” category also includes discharge into rivers. The authors should clarify the distinction between these two classifications.

Authors’ response

Thank you for the comment. In our original submission, the discharge of wastewater into rivers was mistakenly classified as “safe.” We have now corrected this classification. In the revised manuscript, only drainage systems such as flowing sewers, non-flowing sewers/gutters, and permanent pits are classified as “safe,” as these represent designated household disposal channels. All disposal into natural water bodies, including rivers, ponds, lakes, and the sea, is now classified as “unsafe.”

In addition, following this comment, we rechecked all sanitation variables for consistency and identified that the coding order for the water source variable also required correction. We have revised the coding to ensure accurate categorisation, rerun all analyses, and updated Tables 1 and 3 accordingly. The coding has been updated accordingly, the analyses have been re-run, and Table 1 has been revised to reflect the corrected definitions.

We also used this opportunity to recheck all descriptive tables, correct minor typographical errors, and add missing data where previously omitted to ensure the completeness and accuracy of reporting.

Comments from 1st Reviewer #6

Method: Lines 115–134: The explanation of the IFLS dataset is comprehensive; however, the access time (“November 2024”) appears inconsistent with the 2014–2015 data. This should be corrected to avoid the impression that the data were accessed after the analysis had been conducted.

Authors’ response

Thank you for this helpful observation. The date we initially provided referred to when the research team accessed the dataset for this project, not to the IFLS data collection period (2014–2015). To avoid confusion and since this information is not essential to understanding the dataset, we have removed the sentence from the revised manuscript. The Methods section now reports only the official IFLS-5 data collection period.

Comments from 1st Reviewer #7

Method: Ensure consistent use of “Model 1” and “Model 2” or “Model A” and “Model B.” In the Methods section, the text refers to Models 1 and 2, whereas in the Analysis section, it switches to Models A and B.

Authors’ response

Thank you for pointing this out. We have revised the terminology in the Statistical Analysis section to ensure consistency across the manuscript. Specifically, all model has been changed to “Model A” and “Model B” respectively, to match the labels used in the Methods and Results section. In addition, Figure 1 has been revised to reflect these changes, including updating the outcome label from depression to depressive symptoms and explicitly indicating the covariates included in each model, consistent with the SEM specifications described in the Statistical Analysis section.

Comments from 1st Reviewer #8

The discussion remains overly descriptive and does not sufficiently emphasise the study’s novelty compared with previous research. The authors could add a concluding paragraph to Section 4.1 highlighting the contribution of this study relative to research conducted in Bangladesh or Ghana.

Authors’ response

Thank you for the suggestion. We have added a concluding paragraph to Section 4.1 that highlights the contribution of our findings relative to studies from Bangladesh, Ghana, and similar contexts.

Comments from 1st Reviewer #9

Spellings:

1. Lines 231-232: Please use the present tense in the entire result sentence, e.g. "Table 2 presents..."

2. Lines 381-383: The sentence "WHO further estimated that every US$1 invested...", please change to "The WHO further estimates..." in order to parallel the academic style.

3. Use the term “life satisfaction (LS)” consistently throughout the manuscript. Some sections spell it out in full without the abbreviation.

4. Add a source note to each table (e.g., Data source: IFLS-5, authors’ calculation).

5. In British style, the use of a comma before and is not required unless necessary to prevent ambiguity.

6. Line 274: “Standardized coefficient” � “Standardised coefficient”

7. Line 384: “sanitation programs…” � “sanitation programmes…”

8. Please review and ensure consistency with British English conventions across the manuscript.

9. Improve sentence structure for clarity and conciseness, avoiding redundancy.

Authors’ response

We have now implemented all recommended revisions. Specifically, we (1) corrected verb tenses in the Results section, (2) revised the sentence beginning “WHO further estimated…” for consistency, (3) standardised the use of “life satisfaction (LS)” across the manuscript, (4) added source notes to all tables, (5) adjusted comma use in accordance with British English conventions, (6) corrected “standardized” to “standardised,” (7) changed “sanitation programs” to “sanitation programmes,” (8) reviewed the manuscript for full consistency with British English spelling, and (9) improved sentence structure throughout for clarity and conciseness.

Comments from 1st Reviewer #10

Please use the most recent references to support your research. References to numbers [25], [32], [44], [58], and [59] used over the last 10 years.

In example:

Sakti, B., Wibawa, S., Maharani, A. T., Andhikaputra, G., Savira, M., Putri, A., Iswara, A. P., Sapkota, A., Sharma, A., Syafei, A. D., & Wang, Y. (2023). Effects of Ambient Temperature , Relative Humidity , and Precipitation on Diarrhea Incidence in Surabaya. International Journal of Environmental Research and Public Health, 20, 2313.

Wulandari, R., Iswara, A. P., Qadafi, M., Prayogo, W., Astuti, R. D. P., Utami, R. R., Jayanti, M., Awfa, D., Suryawan, I. W. K., Fitria, L., & Andhikaputra, G. (2024). Water pollution and sanitation in Indonesia: a review on water quality, health and environmental impacts, management, and future challenges. Environmental Science and Pollution Research, 0123456789. https://doi.org/10.1007/s11356-024-35567-x

Authors’ response

Thank you for highlighting the importance of using recent references. We have carefully reviewed the citations and updated or supplemented recent references where relevant. For example, we incorporated the recent review by Wulandari et al. (2024) to provide contemporary evidence on water quality and sanitation conditions. However, several references flagged as “older” serve foundational methodological functions and cannot be replaced by more recent studies. Reference (Andresen et al., 1994) provides the original validation of the CES-D–10 scale, which remains the standard reference in both global and Indonesian research. In addition, Amorim et al. (2010) is a core methodological paper that summarises the application of structural equation modelling in epidemiology; more recent SEM literature builds upon, rather than replaces, this foundational work.

Moreover, we have replaced the reference (Guardiola et al., 2014) with a more recent, comparable study (Ammar & Kouser, 2022). Reference (Nelson et al., 2014), which directly examines sanitation conditions and life satisfaction using IFLS data, has been retained because it is uniquely relevant to our study’s focus on WASH factors and subjective well-being in Indonesia. We have therefore updated the references where appropriate and retained essential foundational citations when no more recent or suitable a

---

## [Decision Letter · Decision Letter 1]

14 Jan 2026

Water, sanitation, and depressive symptoms in Indonesia: The mediating role of life satisfaction

PONE-D-25-55524R1

Dear Dr. Maharani,

We’re pleased to inform you that your manuscript has been judged scientifically suitable for publication and will be formally accepted for publication once it meets all outstanding technical requirements.

Kind regards,

Alison Parker

Academic Editor

PLOS One

Additional Editor Comments (optional):

Reviewers' comments:

Reviewer's Responses to Questions

**Comments to the Author**

Reviewer #1: All comments have been addressed

Reviewer #2: All comments have been addressed

Reviewer #3: All comments have been addressed

2. Is the manuscript technically sound, and do the data support the conclusions?

Reviewer #1: Yes

Reviewer #2: Yes

Reviewer #3: Yes

3. Has the statistical analysis been performed appropriately and rigorously?

Reviewer #1: Yes

Reviewer #2: I Don't Know

Reviewer #3: Yes

4. Have the authors made all data underlying the findings in their manuscript fully available?

Reviewer #1: Yes

Reviewer #2: Yes

Reviewer #3: Yes

5. Is the manuscript presented in an intelligible fashion and written in standard English?

Reviewer #1: Yes

Reviewer #2: Yes

Reviewer #3: Yes

Reviewer #1: all the comments has been addressed fairly. I suggest to accept the manuscript and process to the further step

Reviewer #2: The authors have adequately addressed all comments raised during the review process. Revisions were implemented carefully and systematically, resulting in improved clarity, methodological transparency, and coherence across the manuscript. The authors strengthened the rationale, refined variable definitions, clarified analytical approaches, and enhanced the discussion to better align findings with existing literature.

Overall, the revised version reflects a meaningful engagement with reviewer feedback and demonstrates a clear improvement in quality and rigor.

Reviewer #3: Overall, the authors have revised the draft appropriately, and the revisions have clarified all the comments raised during the review.

**Do you want your identity to be public for this peer review?** For information about this choice, including consent withdrawal, please see our Privacy Policy

Reviewer #1: No

Reviewer #2: No

Reviewer #3: No

---

## [Editor Report · Acceptance letter]

PONE-D-25-55524R1

PLOS One

Dear Dr. Maharani,

I'm pleased to inform you that your manuscript has been deemed suitable for publication in PLOS One. Congratulations! Your manuscript is now being handed over to our production team.

Kind regards,

on behalf of

Dr. Alison Parker

Academic Editor

PLOS One